# Cannabis in Hematology Survey Study (CHESS): A Longitudinal Investigation on Uses, Attitudes, and Outcomes of Cannabis Among Hematology Patients Undergoing Hematopoietic Stem Cell Transplant

**DOI:** 10.3390/ijerph22070990

**Published:** 2025-06-23

**Authors:** Andrew I. G. McLennan, Reanne Booker, Cameron Roessner, Marc Kerba

**Affiliations:** 1Department of Psychology, University of Regina, Regina, SK S4S 0A2, Canada; andrew.mclennan@uregina.ca; 2Palliative Care, Arthur J.E. Child Comprehensive Cancer Centre, Alberta Health Services, Calgary, AB T2N 4N2, Canada; reanne.booker@albertahealthservices.ca; 3Pharmacy Services, Arthur J.E. Child Comprehensive Cancer Centre, Alberta Health Services, Calgary, AB T2N 4N2, Canada; cameron.roessner@albertahealthservices.ca; 4Department of Radiation Oncology, Cumming School of Medicine, University of Calgary, Calgary, AB T2N 1N4, Canada

**Keywords:** cannabis, THC, CBD, leukemia, lymphoma, bone-marrow-transplant, alternative therapies, graft-versus-host-disease

## Abstract

Cancer patients use cannabis for medicinal purposes; however, few studies have examined hematology patients’ use of cannabis and no research to our knowledge has investigated the use of cannabis amongst hematology patients before and after hematopoietic stem cell transplant (HCT). The purpose of this longitudinal survey study was to assess aspects of cannabis use in patients who underwent HCT. Eligible patients (*N* = 30) completed two surveys before and 90 days following their HCT. The surveys inquired about several aspects of cannabis (e.g., rate of use, beliefs and attitudes, access to information) and physical and psychological outcomes (e.g., anxiety, comorbidities, graft-versus-host-disease). Rates of cannabis use decreased following HCT (*n* = 14, 46% to *n* = 11, 40%). Conversations on cannabis that were initiated by an oncology health care provider increased post-transplant (*n* = 3, 10% to *n* = 11, 37%). This coincided with fewer who were smoking cannabis as a primary consumption method (*n* = 5, 38 to *n* = 2, 18) and an increase in the use of pharmaceutical cannabinoid products (*n* = 4, 13% to *n* = 6, 21%) as well as oils and topicals. Of the total sample, 63% (*n* = 17) experienced post-treatment complications and 33% (*n* = 10) developed GVHD, six of whom where recent cannabis users. This study provided insight into cannabis use amongst HCT patients and warrants further research with this population, including more exploration of the relationship between GVHD and cannabis.

## 1. Introduction

People living with cancer have been known to utilize cannabis for managing treatment and illness-related symptoms such as pain, nausea, appetite loss, sleep disturbances, and overall well-being [1,2,3,4]. The use of cannabis is well-documented amongst patients with solid-tumor malignancies such as breast cancer and lung cancer [5,6]. However, cannabis use amongst people with hematologic cancers has been studied far less.

Hematopoietic stem cell transplantation (HCT) is offered to patients with hematologic malignancies, and less frequently to those with non-malignant hematologic conditions and auto-immune conditions [7]. Autologous (replacing cells with healthy cells from the host) and allogeneic (replacing cells with healthy donors’ cells) transplants are the two most common options. HCT outcomes are often dependent on a variety of factors, including biomedical (e.g., disease status at the time of transplant, stem cell source), patient-specific (e.g., age, comorbidities), center-specific (e.g., conditioning protocol, prophylaxis), and psychosocial (e.g., coping mechanisms, social support system) factors [8]. Treatment-related morbidity, such as nausea, appetite loss, sleep disturbance, pain, and psychological distress, is often experienced, and these symptoms can persist for months and even years following HCT [9,10]. Graft-versus-host disease (GVHD) remains a leading cause of morbidity and mortality in allogeneic transplant recipients, affecting 30–70% [11,12,13,14]. GVHD can develop acutely post-HCT, generally by day 100, or later as a chronic condition after the initial 100 days of HCT [11,12].

There is some evidence showing that the immunosuppressive and anti-inflammatory components of cannabinoids may reduce incident rates and/or severity of GVHD [15,16,17,18]. There remains uncertainty as to whether cannabinoids attenuate the functioning role of immunocompetent cells and potentially abrogate the graft-versus-leukemia effect, which may contribute to increased risk of relapse [17]. Studies have yet to investigate whether cannabinoids provide comparable alleviation of adverse symptoms in hematology patients compared to that of solid-tumor groups. Moreover, there is little to no research on patient beliefs and attitudes about cannabis in the context of hematological cancer. To our knowledge, no previous studies have investigated the use of cannabis before and following HCT.

The aims of this survey study were to describe past and/or current uses of cannabis amongst HCT patients treated at a tertiary cancer center in Western Canada. We sought to explore patients’ beliefs and opinions related to the use of cannabis and evaluate changes in rates of use, beliefs, and opinions before and following HCT. We also sought to evaluate differences in physical, psychosocial, and emotional well-being between cannabis users and non-users prior to HCT and 90 days following. We selected a 90-day follow up period to evaluate any potential relationship between cannabis use and post-treatment outcomes, such as GVHD.

## 2. Materials and Methods

### 2.1. Participants

Participants were adults with a diagnosis of hematologic malignancy, who were scheduled for and awaiting HCT at the cancer center. Approximately 150 transplants (allogeneic and autologous) are performed at the cancer center annually. All participants signed written consent to participate in this study and for their medical chart information to be reviewed by members of the research team. This study was reviewed and approved by the Health Research Ethics Board of Alberta Cancer (HREBA.CC-19-0461). We ensured that participants’ digital information and privacy and confidentiality were protected using encrypted files and secure data management systems (i.e., RedCap) and that all data was de-identified prior to analysis. Only information pertinent to this study was collected from patient charts by members of the study team.

### 2.2. Measures

#### Pre/Post-HCT Cannabis Surveys

The survey items were adapted from two previously established cannabis-based questionnaires used in oncology populations [2,19], while several new questions were created by the research team that inquired about patients’ experiences, beliefs, and attitudes regarding cannabis. Both surveys were tested with patient partners (*N* = 20) before the study began and their feedback was incorporated into the final versions.

### 2.3. Edmonton Symptom Assessment System—Revised (ESAS-r)

The ESAS-r was included in both surveys to measure clinical symptoms in patients before and after HCT. The ESAS-r is a patient - or provider-reported outcome measure for nine cancer-related symptoms. These symptoms include pain, tiredness/drowsiness, nausea, lack of appetite, shortness of breath, depression, anxiety, well-being, and an additional open response item for other symptoms. Items are scored on a 0-to-10 scale, with higher values for each item reflecting more severe symptoms. A single score per symptom is derived. For instance, on the depression item, 0 = *no depression,* and 10 = *worst possible depression*; and for pain, 0 = *no pain* and 10 = *worst possible pain*. The ESAS-r is one of the most valid and routinely used symptom scales for cancer patients [20,21,22].

#### Fear of COVID Scale (FCV-19S)

The FCV-19S is an assessment instrument that measures a person’s fears toward COVID-19 [23]. The scale consists of seven validated items that inquire about fears and anxiety toward COVID-19. Items are rated on a 5-point Likert scale (i.e., *strongly disagree* to *strongly agree*) where the minimum possible score for each item is 1 and the maximum possible score is 5. Example items on the FCV-19S include (1) *I am most afraid of coronavirus-19*, (2) *it makes me uncomfortable to think about coronavirus-19*, and (3) *I am afraid of losing my life because of coronavirus-19*. Total scores can range from 7 to 35, where a higher total score reflects a higher level of fear. The scale has demonstrated sufficient psychometric properties [23] and has been used to assess levels of fear and anxiety in patients with cancer, including in patients who had undergone HCT during the pandemic [24,25,26].

### 2.4. Procedure

Recruitment began in February 2021 and continued until July 2022. Prospective participants were notified about the study at the time of their first HCT intake appointment by pharmacy staff (facilitated by the co-author CR). Those interested in participating were provided with more information about the study and a secure email link to the digital informed consent form. All study information was distributed and collected on the secure survey database and distribution software, Redcap, (version 9.5) [27]. Once consent was signed, a secure email link to access the first survey was sent to the participant. The second survey was automatically emailed to each participant 90 days following their HCT. Members of the research team (RB and AIGM) extracted and reviewed medical chart data for each patient participant following the completion of both surveys. Clinical variables abstracted from charts included type of transplant (autologous vs. allogeneic), type of hematologic malignancy, treatment-related complications, and comorbidities.

### 2.5. Statistical Methods

Data were summarized using descriptive statistics, including means with standard deviations for continuous variables, and percentages and ranges for categorical variables. The results were organized by each outcome domain per survey, allowing comparisons to be made between both timepoints. We categorized patients as either “recent users” or “non-recent users” of cannabis for the purpose of comparing mean differences between groups on symptom and self-reported outcomes. We considered a “recent user” as anyone who had used cannabis within the past six months and a non-recent user as anyone who had not used cannabis within the past six months. This timeframe aligns with previous research where six months has been used as an indication of recent cannabis use [2]. Group comparisons (i.e., *t*-tests or analysis of variance ANOVAs) were not conducted due to the low sample size and lack of sufficient statistical power.

## 3. Results

### 3.1. Response Rate

Of the 70 patients who were notified about the study, 30 completed both survey one and survey two, for a response rate of 43%. All patients who participated in this study completed both surveys, though two participants did not answer all questions related to their rate of cannabis use.

### 3.2. Patient Demographics and Chart Information

Table 1 presents the demographic characteristics for our sample. Among the respondents, 19 (63%) were male and 11 (37%) were female. The majority identified as Caucasian (*n* = 26, 86%), and most had obtained a post-secondary degree (*n* = 18, 60%). Patient chart data (e.g., cancer type, presence of GVHD, transplant-related complications) are found in Table 2. Notably, we found that 30% of patients overall and 42% of recent cannabis users (i.e., used in the past six months) developed GVHD following treatment.

### 3.3. Past and Current Use of Cannabis

Information on patients’ past and current use of cannabis is displayed in Table 3. Results are described in terms of pre-transplant and post-transplant (90-day follow up) to highlight response differences and similarities between survey one and survey two. Percentages reflect the number of respondents per question, which for some items was not always *N* = 30 because skip logic was used to present or skip certain questions based on the participants’ responses.

**Pre-Transplant**. Regarding cannabis use, 13 (43%) had used some form of cannabis in the past six months and we referred to these respondents as recent users. These individuals were, on average, younger (*M* = 44.00, *SD* = 13.20, Range = 21–64) compared to non-recent or never-before users (*M* = 50.76, *SD* = 17.27, Range = 36–71) and predominantly male (*n* = 8, 73%) (Table 3).

Only participants who indicated that they had used cannabis in the past six months before transplant were asked questions related to their current use of cannabis. Among the 13 respondents, 9 (69%) reported using cannabis daily or almost daily (Table 3).

**Post-Transplant**. Eleven (36%) indicated that they had used cannabis in the past three months (post-transplant), with six of these participants (54%) using it daily, and three (27%) on a weekly to monthly basis. Six (54%) were using a high-THC concentrated strain (Table 3).

### 3.4. Reasons for Using Cannabis

**Pre-Transplant**. Table 4 presents information on attitudes towards reasons why all patients believed cannabis could be or is useful. Most (*n* = 21, 70%) believed it could help improve sleep, relieve cancer/treatment-related pain (*n* = 20, 66%), and nausea (*n* = 16, 53%). Only three (10%) reported that cannabis would not be helpful for any of the stated reasons.

**Post-Transplant**. Reasons for using cannabis remained similar post-HTC; however, fewer (*n* = 16, 53%) reported that it would be helpful for sleep, pain (*n* = 15, 50%), and appetite or weight gain (*n* = 11, 36%), while more reported that it would be useful for nausea (*n* = 13, 43%) and quality of life (*n* = 8, 26%). After three months, most were still accessing cannabis from a cannabis retailer (*n* = 10, 33%); however, some began acquiring medically prescribed cannabis as an alternative.

### 3.5. Information Seeking

**Pre-Transplant**. Most patients accessed information from online websites (*n* = 18, 60%), family and/or friends (*n* = 13, 43%), or from books and magazines (*n* = 9, 30%). Only five (16%) had sought out information from a medical cannabis practitioner or an oncologist. The majority (*n* = 27, 90%) reported that their oncologist or oncology health care provider had not initiated a conversation with them about cannabis use (Table 4).

**Post-Transplant**. Following transplant, respondents were continuing to access most of their information from the same sources; however, there was a rise in those who sought out information from a medical cannabis practitioner (*n* = 5, 16%) or their oncologist (*n* = 7, 23%). Most (*n* = 19, 63%) still had not been approached by an oncology health care provider to discuss cannabis use; however, there was an increase from pre-transplant (*n* = 11, 37%). Comparisons between pre and post treatment responses for select variables are shown in Appendix A. 

### 3.6. Physical, Psychosocial, and Emotional Well-Being

Table 5 displays the means, standard deviations, and ranges for scores from the ESAS-r at both timepoints. We observed a decrease in adverse symptoms reported following HCT. Recent users of cannabis reported higher mean symptoms than others following transplant (*M* = 29.13, *SD* = 7.68, Range = 17–42; *M* = 16.00, *SD* = 6.15, Range = 9–30).

### 3.7. Cannabis and Perceptions of COVID-19

Means and SDs for the FCV (Fear of COVID Score)-19S are found in Table 5. Six patients (20%) had either begun or increased their use of cannabis since the start of the pandemic, and a large proportion of pre-transplant users (*n* = 8, 32%) were using both for medical and recreational purposes. Recent users also reported higher FVC-19S scores compared to non-recent users following transplant (*M* = 17.70, *SD* = 5.43, Range = 7–28 versus *M* = 15.29, *SD* = 5.20, Range = 7–23).

## 4. Discussion

### 4.1. Overview of Findings

In our Canadian study sample, cannabis was used at a rate that was comparable to solid-tumor groups [5]. This was a unique investigation in that data were collected before and following HCT, which to our knowledge has not been previously reported in the literature. Our findings showed that hematology patients were using cannabis at a relatively high rate and believed that it is helpful for managing cancer symptoms, with most accessing cannabis and cannabis-based information from informal sources. We observed a relatively high rate of GVHD post-treatment in patients who were recent users of cannabis, which may be attributed to other variables that were not measured in this study (e.g., treatment complications, baseline health status, disease severity). However, it may also be that patients who developed GVHD were more symptomatic, felt more unwell, and were using cannabis to manage their symptoms, which may also explain the higher mean adverse symptoms scores before and after transplant for recent cannabis users. Our findings substantiate the need for further study to better understand the use and implications of cannabis in hematology.

### 4.2. Patient Chart Information

Most patients were diagnosed with leukemia or lymphoma and underwent allogeneic HCT. There were considerable comorbid chronic health concerns, such as hypertension and respiratory conditions. Over half of the sample reported post-transplant complications such as GVHD, serious infections, and immune deficiency. Participants also reported that they experienced problems with mood (e.g., anxiety, depression), insomnia, well-being, and pain following HCT, which are commonly reported problems associated with this treatment [1,28]. The development of GVHD occurred in one-third of the sample following HCT. Amongst the sample, 42% (6/14 total) of recent cannabis users (within the past six months) developed GVHD following transplant, which is a curious finding given evidence in past research that cannabis may alleviate the intensity or development of GVHD [15,16]. The immunomodulatory and anti-inflammatory properties of CBD [29,30] are of particular interest in the context of GVHD, where alloreactive donor T cells recognize host tissues as foreign and mount an immune response [31,32]. Murine models have found that CBD reduces T cell responses and is associated with less GVHD [32,33]. Yeshurun et al. (2015) [16] conducted a Phase 2 trial to examine CBD in addition to cyclosporine and methotrexate as prophylaxis for GVHD in patients who underwent allogeneic HCT (*N* = 48). The authors found that none of the patients who used CBD developed GVHD. Compared to historical controls, the authors reported a hazard ratio of 0.3 (*p* = 0.0002) for developing grade II to IV acute GVHD for patients treated with CBD in addition to standard GVHD prophylaxis, and for patients who survived beyond 100 days, the cumulative incidences of moderate-to-severe chronic GVHD at 12 and 18 months were 20% and 33%, respectively. Further exploring the biological mechanisms of cannabis and GVHD is necessary to fully understand this relationship.

### 4.3. Past and Current Use of Cannabis and Reasons for Use

Findings indicated that a considerable number (nearly half) of patients had used cannabis in the past six months before HCT and most had been using on a daily or weekly basis. Frequency of cannabis use decreased following transplant, which may have been in part due to the alleviation of adverse symptoms, or by recommendation from their hematology team. Cannabis use ranges from 20% to nearly 50% amongst people living with cancer according to current evidence in the literature [1,2,34], and a large-scale survey study (*N* = 612) reported that 42% (*n* = 257) were using cannabis for medical purposes related to their cancer [35]. However, most findings are reported on solid-tumor groups, with far less examining hematology populations. A single study that set out to examine patterns of cannabis use in patients with Hodgkin Lymphoma (*N* = 133) found that 38% (*n* = 51) of patients were using cannabis, with the majority reporting improvements in general well-being (87%, *n* = 44), appetite (82%, *n* = 41), and nausea (78%, *n* = 39) [36]. Evaluating cannabis use and changes in use during active treatment is also less studied. A recent survey study of cannabis consumption patterns among cancer patients in the United States (*N* = 385) found that cannabis use reduced by 30% during cancer treatment and that use during treatment was lowest among patients with blood cancers (2.53%) compared to solid-tumor groups like breast (16.62%) and prostate (8.31%) cancer [37]. Our findings are similar in that fewer patients were using cannabis post-HCT.

Most patients in our study switched their method of consumption from smoking to ingested oils and extracts or edibles following transplant, which may be attributed to education from hematology HCPs as it is common practice at this site to recommend ingestion and vaporization over smoking. Shifts in method of consumption throughout treatment are not commonly reported as most research is cross-sectional in design. Current evidence shows that between one-third to over half of patients use inhalation (i.e., smoking dry flower or vaping) as their preferred method of consumption. In contrast to our findings, one survey study of cancer patients’ use of cannabis before and during treatment (n = 185) reported that inhalation/smoking increased during cancer treatment (54% to 60%) compared to pre-treatment [37]. Given our findings that more patients were discussing their use of cannabis with their hematology team after HCT, the switch to a healthier alternative consumption method (i.e., oils, extracts, edibles) may have resulted from expert guidance.

Among recent users of cannabis, most were using it to relieve symptoms of pain, insomnia, and fatigue, and to improve overall well-being. These findings are congruent with previous work suggesting that patients are using cannabis for a variety of cancer-related adverse symptoms [2,35,36]. A recent systematic review [5] examined modes, patterns, reasons, discontinuation, and adverse experiences of cannabis use in patients with cancer. The authors included 27 studies in their review and reported that the main motivation for patients to use cannabis was symptom management due to cancer or its treatment, including pain, nausea, insomnia, and anxiety. Most participants across the studies found that cannabis improved their symptoms. The authors suggested that more longitudinal studies should be conducted to assess the positive experiences with cannabis use compared to any adverse experiences that may occur in this vulnerable population. Our findings are unique in that they highlighted that patients were using cannabis to manage symptoms that are most prevalent following HCT. A Canadian cohort study (2006–2017) that investigated adverse symptoms following HCT (*N* = 5844) reported higher rates of moderate-to-severe levels of pain, drowsiness, general distress, anxiety, and depression symptoms following treatment [38]. We found similarly that our patient sample experienced more severe pain, drowsiness, anxiety and depression symptoms, and lower overall well-being, and that patients overall felt that cannabis would be helpful for many of these symptoms.

### 4.4. Access and Information Seeking

Our results showed that most patients were accessing information on cannabis from less evidence-based resources, such as websites, social media, and family and friends. There is reason for concern when people undergoing cancer treatment access cannabis-based information from less-valid sources given the lack of evidence-based guidance this offers to people. This likely reflects the blurred lines between recreational and medical uses of cannabis, especially given that only a few of our participants had a prescription for a Health Canada-approved pharmaceutical cannabinoid product.

Compared to the pre-transplant setting, we observed a large shift in the proportion of patients who received information or guidance about cannabis from health care professionals (pharmacists and oncologists) after 90 days post-transplant. This finding is of interest given that recent evidence shows that health care providers are less likely to initiate conversations with their patients about using cannabis [2,19, 39] and health care providers have often reported not feeling confident in their professional knowledge on the use or prescription of cannabis for patients [39,40,41]. In the present study, most patients reported that an oncology health care provider had still not initiated a conversation about using cannabis at pre-treatment, which is consistent with previous reports [41,42] and reinforces the need for continued training and education on cannabis for medical trainees and oncology HCPs.

### 4.5. Physical and Psychological Symptoms

Unwanted physical and psychological symptoms were present irrespective of cannabis use and these scores were lower following HCT. More recent cannabis users experienced adverse symptoms than non-users. These findings may be explained by various health- and treatment-related factors, such as the type of hematologic cancer, treatment doses of chemotherapy and other treatment prior to HCT, and other comorbid health conditions that may have contributed to psychological and physical comorbidity. It may also be that patients who were more symptomatic and unwell before treatment were using cannabis to manage their symptoms.

It is equally important to consider the potential adverse effects of cannabis use, including side effects as well as possible impacts on outcomes such as progression-free survival (PFS) and overall survival (OS). For example, a recent study [43] evaluated the impact of cannabis use on clinical outcomes on patients with solid tumors receiving immuno-oncology treatment. The authors conducted a retrospective cohort study that included 106 patients and found that median OS in cannabis users was 6.7 months compared to 17.3 months in non-users (HR, 1.78; 95% CI, 1.06–2.97; *p*  =  0.029). The median PFS was 4.8 months in cannabis users compared to 9.7 months in non-users (HR, 1.74; 95% CI, 1.09–2.79; *p*  =  0.021).

### 4.6. Clinical Implications and Recommendations

Cannabis has potential as an adjunct to traditional treatment, yet more work is needed to understand how it best fits within the hematology cancer care framework. Our findings suggest that more research is needed to fully understand the relationship between GVHD development and cannabis use. While we observed a higher rate of GVHD in recent cannabis users, there is evidence suggesting that cannabis may in fact reduce the risk of developing GVHD [16]. Studying this relationship across other samples undergoing HCT, as well as more attention to the underlying biological mechanisms, is needed. Our findings also showed that patients shifted to healthier consumption methods post treatment, which also coincided with more patients who were talking to their oncologist about their cannabis use. Clinicians should be sure to initiate open conversations with their patients about cannabis, including questions on how they use it, how often, the constituents (THC, CBD) and dosing.

The recent American Society of Clinical Oncology (ASCO) clinical practice guidelines on cannabis and cancer are a useful resource for oncology clinicians who are seeking information on cannabis and cancer (e.g., clinical practice recommendations, cannabis-based professional resources, up-to-date empirical evidence on cannabis in cancer) [4]. These expert guidelines recommend that oncology health care providers routinely and non-judgmentally inquire with patients about cannabis use [4]. They also recommend that oncology health care providers have sufficient knowledge of the empirical evidence on cannabis to communicate and offer helpful suggestions to their patients [4]. The ASCO guidelines emphasize that cannabis should not be recommended for cancer-directed treatment except for in the context of clinical trial participation. In addition, the guidelines suggest that cannabis might be helpful for refractory, chemotherapy-induced nausea and vomiting if added to guideline-concordant antiemetic regimens [4]. Health care staff should be careful to help patients weigh the potential risks and benefits of using cannabis during treatment, while researchers should look to evaluate the short- and long-term health risks of cannabis use, especially in patients undergoing intensive treatment like HCT.

### 4.7. Limitations

There are several important limitations worth noting. Our sample size was smaller than anticipated and we were unable to conduct more complex analyses (i.e., analysis of variance ANOVA; linear regression) due to a lack of statistical power. This was due to COVID-19 restrictions, resulting in fewer HCT treatment appointments overall and thus eliminating the option of in-person recruitment. Nevertheless, we did recruit a reasonable sample size among those patients who had been informed about the study. We also had a considerably high number of respondents who were recent cannabis users and less who were non-recent users. Patients who elected to participate in the survey may have held a favorable implicit bias toward cannabis, which may have motivated participation. However, the rate of use reported in our study is still consistent with other reports. We did not inquire about adverse effects of cannabis in this study, nor did we inquire about long-term adverse outcomes; documenting these outcomes is important given the potential for contaminants and subsequent increased risk of infection in this population. Finally, given the use of a single recruitment site and our sample size, we recognize that our findings may be difficult to generalize to the broader hematology patient population.

## 5. Conclusions

This study found that hematology patients undergoing HCT are using cannabis at a high rate and believe that it is helpful for managing treatment-related symptoms. This evidence also contributes to our ongoing understanding of the relationship between cannabis and treatment-related outcomes, such as GVHD. While the use of cannabis has increased in recent years, the research on cannabis use in patients with cancer remains limited to date. More studies are needed to evaluate the potential benefits, risks, and therapeutic potential for patients with cancer, including those with hematologic malignancies and those undergoing HCT. In the meantime, clinicians should be prepared to answer questions pertaining to cannabis use, provide information on the known potential risks and benefits associated with cannabis use in the context of cancer, and assist patients in the safe usage of cannabis.

## Figures and Tables

**Table 1 ijerph-22-00990-t001:** Participant characteristics.

Characteristics	Value	
	Mean (*SD*) [Range]	
Age (years)	47.80 (16.2) [21–71]	
	** *N* **	**%**
Gender		
Female	11	37
Male	19	63
Non-Binary/Other	0	0
Race/Ethnicity		
First Nations/Indigenous	2	6
Asian	2	6
African American/Black	0	0
Latino	0	0
Caucasian/White	26	86
Other	0	0
Marital Status		
Single	7	23
In a relationship (not married)	3	10
Married	17	56
Widowed	0	0
Divorced	1	3
Other	2	6
Education		
Elementary to Highschool	0	0
Trades/post-secondary diploma	8	26
University (undergraduate)	10	33
University (graduate/professional/post-doctoral)	6	20
Other	3	10
Diagnostic Information		
Lymphoma	12	40
Leukemia	12	40
Other cancer type	5	16

**Table 2 ijerph-22-00990-t002:** Patient chart information.

Item	*N* (%)
Cancer Type	
Leukemia	11 (36)
Lymphoma	11 (36)
Multiple Myeloma	1 (3)
Other	5 (17)
Type of Transplant	
Allogeneic	17 (56)
Autologous	11 (36)
Presence of Graft Versus Host Disease (GVHD)	
Yes	10 (33)
No	17 (66)
Transplant-Related Complications	
Yes	17 (63)
No	11 (37)
Presence of Adverse Symptoms	
Psychological/Mood	9 (32)
Respiratory	5 (18)
Hypertension	5 (18)
Sleep Issues	2 (7)
Pain (chronic or acute)	8 (28)
Chronic Health Conditions	15 (53)
Kidney/Bladder Infection	5 (18)
Gastroesophageal Reflux Disease	5 (18)
Cancer-Specific Comorbidity	2 (7)
Autoimmune	1 (3)
COVID-19	
Yes	2 (7)
No	26 (92)
Total	28

**Table 3 ijerph-22-00990-t003:** Cannabis use characteristics (pre/post-transplant).

Characteristics	Pre-Transplant (Past 6 Months)	Post-Transplant (Past 3 Months)
	*N* (%)	*N* (%)
Have you used cannabis in any form in the last six months?		Over the past three months
Yes	14 (46)	11 (40)
No	16 (54)	17 (60)
Do you currently hold a prescription for medical cannabis?		
Yes	3 (10)	3 (10)
No	27 (90)	25 (90)
Are you currently using, or have you previously used, a Health Canada approved pharmaceutical cannabinoid product? (e.g., nabilone, dronabinol, etc.)		
Yes	4 (13)	6 (21)
No	26 (86)	22 (78)
When was the last time you used cannabis?		
Less than 1 week ago	11 (36)	10 (37)
More than 1 week but less than 6 months	2 (6)	1 (10)
More than 6 months but less than 5 years	4 (13)	3 (11)
More than 5 years	4 (13)	5 (19)
I have not used cannabis	8 (27)	6 (22)
How often have you used cannabis products in the last six months?	N = 13	N = 11
Daily or almost daily	9 (81)	6 (54)
Weekly	3 (23)	2 (18)
Monthly	1 (7)	2 (18)
Less than 6 times in past six months	0 (0)	2 (18)
If you are aware, what type of strains of cannabis/cannabis extracts do you prefer to use? (I prefer strains with…)		
High THC concentration	6 (46)	5 (45)
High CBD concentration	2 (15)	5 (45)
Hybrid of THC and CBD	3 (23)	1 (9)
I do not know	3 (23)	1 (9)
How do you most commonly use cannabis? (Please select the option that best applies to you)		Over the past three months
Smoked as a cigarette (joint), through a pipe, or another device)	5 (38)	2 (18)
Ingested oils/extracts	4 (30)	5 (45)
Consumed as a cooked recipe or in liquid form	0 (0)	1 (9)
Electric vaporizer	2 (15)	2 (18)
Oral spray	1 (7)	0 (0)
Topicals	0 (0)	0 (0)
Other (please specify)	1 (7)	2 (18)
In an average month, how much would you typically spend on cannabis?		In an average month, over the past three months
CAD 10 to CAD 49	7 (53)	3 (27)
CAD 50 to CAD 99	0 (0)	4 (36)
CAD 100 to CAD 199	2 (15)	2 (18)
CAD 200 to CAD 499	3 (23)	1 (9)
CAD 500 to CAD 999	1 (7)	0 (0)
How do you most frequently acquire cannabis?		
Family/friends	7 (23)	2 (6)
Online from a licensed provider (LP)	5 (16)	4 (13)
Cannabis retailer	12 (40)	10 (33)
Doctor (oncologist/another physician)	1 (3)	1 (3)
Pharmacist	0 (0)	2 (6)
I do not acquire my own cannabis	11 (36)	10 (30)
Other	1 (3)	2 (6)

Note. Calculations were made using a denominator of *N* = 30 unless specific otherwise.

**Table 4 ijerph-22-00990-t004:** Beliefs and information seeking (pre/post-transplant).

Characteristics	Pre-Transplant (Past Six Months)	Post-Transplant (Past Three Months)
	*N* (%)	*N* (%)
Please select all of your reasons for using cannabis, or reasons why you think cannabis could be used		
To improve quality of life	12(40)	8 (26)
To treat myself	7 (23)	6 (1)
To cure my cancer	3 (10)	3 (10)
To relieve cancer/treatment pain	20 (66)	15 (50)
To help me sleep	21 (70)	16 (53)
To relieve sadness/depression from cancer and treatment	13 (43)	8 (26)
To reduce nausea/vomiting	16 (53)	13 (43)
To improve appetite/weight gain	12 (40)	11 (36)
For recreational use/curiosity	2 (6)	9 (30)
I don’t believe cannabis can be used for any for these reasons	3 (10)	2 (6)
Do you believe that cannabis has or could help manage/treat any of the following symptoms?		
Cancer-related pain	19 (63)	20 (71)
Fatigue	8 (26)	7 (25)
Nausea/vomiting	17 (56)	15 (54)
Appetite loss	17 (56)	18 (63)
Anxiety, depression, general sadness	18 (63)	15 (53)
Sleep	21 (70)	21 (75)
No, cannabis is not or has not been helpful for treating any of my symptoms	3 (10)	21 (75)
From what sources have you acquired information on cannabis?		
Online websites	18 (60)	7 (23)
Social media sites	4 (13)	14 (46)
Friends and/or family	13 (43)	6 (20)
Books and/or magazines	9 (30)	6 (20)
Medical cannabis practitioner	3 (10)	5 (16)
Oncologist	2 (6)	7 (23)
I have not acquired any information on cannabis	10 (33)	10 (33)
Has an oncologist or oncology health care provider ever initiated a conversation with you about cannabis?		
Yes	3 (10)	11 (37)
No	27 (90)	19 (63)

**Table 5 ijerph-22-00990-t005:** Physical and psychological outcomes (pre/post-transplant).

Measure	Pre-Transplant (Past Six Months)		Post-Transplant (Past Three Months)	
	Mean Value (*SD*) [Range]			
Edmonton Symptoms Assessment System—Revised (ESAS-r; total sample)	24.5 (12.9) [9–66]		22.6 (9.7) [9–42]	
ESAS-r Symptoms				
Pain	2.68 (2.30) [1–10]		2.33 (1.95) [1–8]	
Tiredness	3.84 (2.10) [1–8]		3.33 (2.06) [1–7]	
Drowsiness	3.12 (2.13) [1–8]		2.73 (1.94) [1–6]	
Nausea	1.76 (1.45) [1–6]		1.87 (1.30) [1–4]	
Appetite	1.8 (1.35) [1–5]		2.46 (1.76) [1–5]	
Shortness of Breath	1.68 (1.68) [1–6]		2.13 (1.74) [1–6]	
Depression	2.36 (2.16) [1–10]		2.20 (1.97) [1–6]	
Anxiety	3.16 (2.43) [1–8]		2.6 (2.47) [1–8]	
Well-being	4.08 (2.51) [1–8]		3.33 (2.06) [1–10]	
ESAS-r (recent vs. non-recent)	Recent User	Non-Recent	Recent User	Non-Recent
	30.2 (14.1) [18–66]	23.7 (11.4) [9–47]	29.5 (7.7) [17–42]	16 (6.1) [9–30]
FCV-19S (total sample)	14.7 (5.7) [7–31]		16.2 (5.3) [7–28]	
FCV-19S (recent vs. non-recent)	Recent User	Non-Recent	Recent User	Non-Recent
	16.6 (4.8) [7–22]	13.3 (6.0) [7–31]	17.7 (5.4) [7–28]	15.3 (5.2) [7–23]

## Data Availability

The data presented in this study are available on request from the corresponding author to limit risks to confidentiality, given that this was a small sample of patients with hematologic cancer from the same local cancer center.

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
