# Peer review of "Cannabis in Hematology Survey Study (CHESS): A Longitudinal Investigation on Uses, Attitudes, and Outcomes of Cannabis Among Hematology Patients Undergoing Hematopoietic Stem Cell Transplant"

_ijerph, 2025, doi:10.3390/ijerph22070990_

Round 1

Reviewer 1 Report

Comments and Suggestions for Authors

Line 145: 90-day follow up is better grammar.
Line 209: due to the small patient numbers, this statement is too strong. In addition, it could be that those who had treatment related adverse events were more likely to use Cannabis, not the other way round that Cannabis increased their chance of having adverse events. It may be the reason why it does not agree with the literature. It was mentioned above that cannabis users tend to be younger.
Line 222: please quote the absolute patient numbers, readers or reviewers do not want to scroll up to look for that in the result, ie. 40% (xx/yy total)

Reviewer 2 Report

Comments and Suggestions for Authors

This manuscript presents a novel longitudinal investigation of cannabis use among hematology patients undergoing hematopoietic stem cell transplant (HCT), a significantly understudied population. Key strengths include its unique pre- and post-HCT assessment of cannabis use patterns, patient beliefs, and clinical outcomes, revealing high utilization rates (46%) and notable associations between frequent cannabis use and increased GVHD incidence—a finding that warrants further mechanistic exploration. The study also highlights critical gaps in patient-provider communication regarding cannabis, emphasizing the need for improved clinician education and evidence-based guidance in this vulnerable population.

#1 The specific content and scoring criteria of the survey instruments (e.g., ESAS-r, FCV-19S) are not described in detail in the text, and it is not clear how to define “recent user” and “non-recent user”. In the “Materials and Methods” section, add specific entries and scoring methods for the survey instrument. Clearly define the time frame (e.g., past 6 months) for “recent user” and describe the basis for categorization.

The #2 study found a higher incidence of GVHD among cannabis users, but did not discuss possible biological mechanisms. Cite additional literature (PMID: 39811936) in the Discussion section that explores potential mechanisms of cannabinoids and immune regulation.

#3 The discussion does not adequately compare existing studies (e.g., solid tumour vs. hematology patients) and has insufficient recommendations to guide clinical practice. Add comparisons with similar studies to highlight the uniqueness of this study (e.g., HCT patient population). Make specific clinical recommendations, such as how HCPs should discuss cannabis use with patients.

#4 Some abbreviations (e.g., HCT, GVHD) are not fully labeled on first occurrence and the denominator is not consistently stated for “N (%)” in the table. Ensure that all abbreviations are labeled in full when they first appear. Indicate the denominator for percentage calculations (e.g., “N=30”) in the table title or footnote.

Given these considerations, I highly recommend that authors revise their manuscript. Looking forward to receiving your revised version of the manuscript. I will review this manuscript again based on the revised version.

Reviewer 3 Report

Comments and Suggestions for Authors

The specific content of ethical review was not mentioned, although it was mentioned that the research passed ethical review, it did not provide detailed instructions on how to protect participant privacy or handle sensitive data.

The statistical method is simple, using only descriptive statistics without conducting more complex analyses (such as regression analysis), which limits the depth of the results.

Long term effects were not discussed, and the study only followed up for 90 days without evaluating the long-term effects or safety of Cannabis use.

The lack of a control group that did not use Cannabis in the study makes it difficult to clarify the causal relationship between Cannabis use and symptom improvement or GVHD.

The study did not adequately control for confounding variables that may affect the results, such as the patient's other medications, treatment dosage, or baseline health status.

The study did not provide detailed records of the specific dosage, frequency of use, or composition of Cannabis, which limited the interpretation of the results.

The relationship between GVHD and cannabis use is unclear. Research has found that cannabis users have a higher incidence of GVHD, but the mechanism or causal relationship has not been thoroughly explored, only suggesting the need for further research.

Please consider the above issues and provide responses and necessary modification.

Round 2

Reviewer 3 Report

Comments and Suggestions for Authors
  1. The list of Abbreviations does not include all the Abbreviations in the text.
  2. The abstract should be more quantitative, which can include some key data in the following studies.
  3. The table caption should be above the three-line format.
  4. At least one figure is suggested in the manuscript.
